ecology/applied mathematics/theoretical biology

critical slowing down, complex networks, stability, critical transitions

**Author for correspondence:**
Amin Ghadami
e-mail: aghadami@umich.edu

# Data-driven identification of reliable sensor species to predict regime shifts in ecological networks

Amin Ghadami, Shiyang Chen and Bogdan I. Epureanu

Department of Mechanical Engineering, University of Michigan, Ann Arbor, MI, USA

AG, 0000-0001-5883-4153; SC, 0000-0001-8974-3705

Signals of critical slowing down are useful for predicting impending transitions in ecosystems. However, in a system with complex interacting components not all components provide the same quality of information to detect system-wide transitions. Identifying the best indicator species in complex ecosystems is a challenging task when a model of the system is not available. In this paper, we propose a data-driven approach to rank the elements of a spatially distributed ecosystem based on their reliability in providing early-warning signals of critical transitions. The proposed method is rooted in experimental modal analysis techniques traditionally used to identify structural dynamical systems. We show that one could use natural system fluctuations and the system responses to small perturbations to reveal the slowest direction of the system dynamics and identify indicator regions that are best suited for detecting abrupt transitions in a network of interacting components. The approach is applied to several ecosystems to demonstrate how it successfully ranks regions based on their reliability to provide early-warning signals of regime shifts. The significance of identifying the indicator species and the challenges associated with ranking nodes in networks of interacting components are also discussed.

## 1. Introduction

Complex systems might undergo abrupt transitions from one stable state to another [1]. Rapid degradation of vegetation cover [2], sudden collapse of global financial markets [3], failures in large-scale electric power transmission systems [4], abrupt climate change [5,6] and collapse of populations in ecosystems [7,8] are examples of such dramatic changes. Hence, anticipating such abrupt transitions is of great importance as part of a preventive plan against possible detrimental consequences.

The complexity of the underlying configuration of the systems and the interactions between elements makes it difficult to develop detailed and accurate models to predict when critical transitions might occur. To overcome this challenge, researchers have focused on developing model-free approaches to detect early-warning signals of impending critical transitions [9–12]. Most existing approaches are based on the phenomenon of critical slowing down, which is defined as the slowing down of the dynamics around an equilibrium when a system approaches a tipping point [13,14]. In particular, early-warning signals are statistical indicators that may reveal proximity to a tipping point based on slowing-down phenomenon. These signals have been developed for systems with small fluctuations around their equilibrium state resulting from stochastic perturbations. As a system approaches a bifurcation that exhibits critical slowing down, the rate at which the system recovers from perturbations decreases, and the time required for the system to return to its equilibrium state increases. Thus, the dynamics becomes more correlated with its past, which leads to an increase in autocorrelation. Furthermore, perturbations can accumulate, which leads to an increase in the size of the fluctuations and as a result, an increase in variance [8]. Some manifestations of critical slowing down, i.e. increasing variance and autocorrelation, in the temporal dynamics of a system have been proposed as model-free early-warning signals to anticipate critical transitions [9,11,15,16].

The advantages of applying early-warning signals to successfully predict when a system is approaching a tipping point have been investigated in recent studies [11,16–23]. However, the application of such early-warning signals to large-dimensional systems (such as spatially distributed systems with many components), remains a major challenge. The main challenging attributes of such systems are their complex dynamics involving large numbers of interacting components. The utility of the information collected from system components in regard to critical slowing down depends on both the topology and dynamics of the system in question. Recent studies have shown that not all system components provide the same signal strength for predicting upcoming transitions, some components show a clear warning sign of impending transitions but others are unable to detect changes in the system dynamics [24–26]. Change in the correlation and connectivity pattern has also been shown to be more significant among a subset of system components by approaching to a transition [27–29]. Due to practical constraints, however, it is often infeasible to measure the dynamics of all components of a system at all times to detect these early-warning signals. Measuring only a subset of system components, on the other hand, may lead to critical transitions remaining undetected, because the measured components may not provide a strong enough signal to detect the upcoming transition. Thus, determining which system components to monitor and how to interpret the measurements to obtain accurate early-warning signals is an important consideration.

Critical slowing down, which is the basis for detecting critical transitions, is associated with the dominant eigenvalue of the locally linearized system suggesting that the states most closely aligned with the corresponding dominant eigenvector are among the best candidates for measurements to extract early-warning signals. Mathematically, this is because the dynamics along the direction of the dominant eigenvector are at their slowest as the system approaches the critical transition. It is challenging or even not feasible, however, to estimate the dominant eigenvector of a complex system in the absence of an accurate model. The dominant eigenvector of a covariance matrix generated from system measurements has been proposed as an approximation of the dominant direction in the system dynamics [30]. However, the efficiency of such a variance-based approach is decreased when the system is subject to heterogeneous measurement and process noise.

Due to the costly consequences of stability loss and critical transitions in complex systems, new methods which improve interpreting the recorded data and increase the reliability of the predictions are demanding. Herein, we introduce a data-driven approach to rank the species in an ecosystem (or the regions of a spatially distributed ecosystem) based on their reliability in providing strong early-warning signals of critical transitions. For this purpose, a data-driven algorithm, known as the eigensystem realization algorithm, is employed to approximate the eigenvectors of systems, and as a result, to identify the best system components to monitor to extract the most reliable early-warning signals of critical transitions. The system dynamics in response to natural stochastic excitations is used as input to the algorithm to approximate its dynamical properties. This approach is capable to identify the eigenvectors of a system with heterogeneous stochasticity. In addition, the proposed approach identifies several eigenvectors, not just the slowest one, which makes it a great candidate for future research focusing on post-processing measurements. Several numerical examples are discussed and the species in the considered ecosystems are ranked based on their capability to provide adequate early-warning signals.

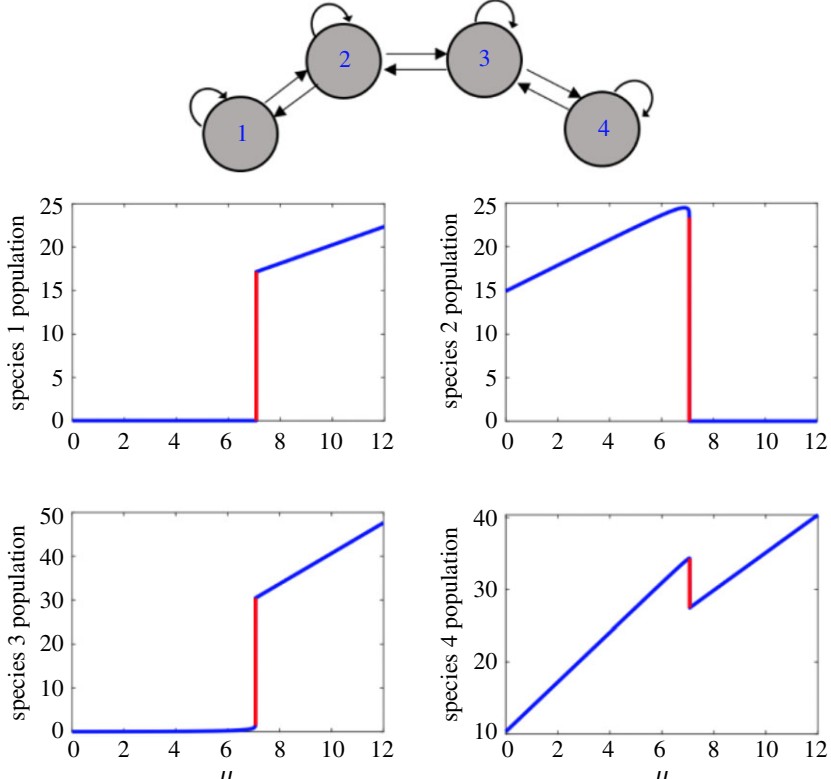

**Figure 1.** Bifurcation diagrams for each node in a network of four interacting species in a chain configuration. A critical transition occurs at $\mu = 6.8$.

# 2. Significance of identifying the slowest direction and the best indicator species

Consider a network of four connected species in a chain configuration (figure 1). The dynamics of each species are governed by the following stochastic differential equation:

$$dx_i = \left[ r_i x_i \left( 1 - \frac{\sum_j \alpha_{ij} x_j}{K_i} \right) + u_i \right] dt + \sigma_i dW_i, \qquad i,j = 1,2,3,4, \tag{2.1}$$

where $r_i$ is the maximum intrinsic growth rate, $K_i$ is the carrying capacity of species $x_i$, and the competition between species is defined by the coefficient $\alpha_{ij}$. The values for $r_i$, $K_i$ and $\alpha_{ij}$ are randomly chosen from the intervals [0.6,1], [5,15] and [0,1.5], respectively. Moreover, a small immigration term $u_i$ is assumed in the population of each species to mimic dispersal that prevents species from reaching extinction and negative values [30,31]. Parameter $\mu$ is used as control parameter following a previous study [31] to reflect the effect of environmental change on the system by modifying the carrying capacity of each species as $K_i' = K_i(1 + \eta_i \mu)$. Parameters $\eta_i$ are selected randomly from the interval [0,1] for each species, reflecting the fact that the change in the environment does not affect all species in the same way. By increasing parameter $\mu$, the system undergoes a critical transition (figure 1). In equation (2.1), $\sigma_i$ is the noise level for state variable $x_i$. Note that the stochasticity can shift the critical point, and therefore the critical transition may be observed before the deterministic critical value. Strong stochasticity in the system dynamics has more significant effect on the stability diagrams and the predictability of transitions. To showcase our methods, we focus on small stochasticity in the equations so that the risk of transitions well before the critical point is low and early-warning indicators can increase to their maximum before transition.

By varying the control parameter from $\mu_{\text{start}} = 3.5$ to $\mu_{\text{end}} = 6.8$, the model in equation (2.1) is numerically integrated to obtain surrogate measurements taken from each node. Heterogeneous random excitations are modelled as independent random walk processes, and were added to the

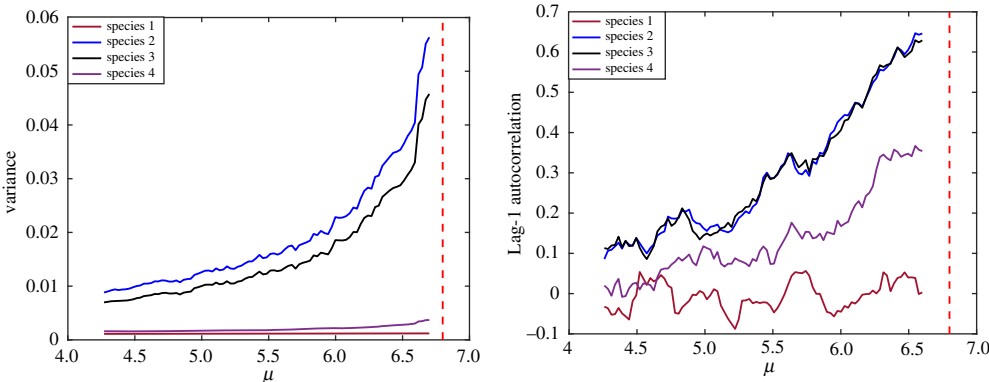

**Figure 2.** Early-warning signals recorded using each node in a network of four interacting species in a chain configuration. Species 2 and 3 provided the most significant early-warning signals of the critical transition. The vertical dashed line represents the critical transition.

dynamics. Early-warning signals of critical transition, namely variance and autocorrelation, estimated from measured time series of the state variable $x_i$ using a sliding (overlapping) moving window (figure 2). It is observed that species 2 and 3 show the most significant signal of approaching a transition, while species 1 and 4 do not provide much useful information and show such a small increase which becomes easily affected by uncertainties in the measurements and data processing.

To perform a more detailed analysis of the warning indicators provided by each species, this procedure was repeated 100 times with random heterogeneous measurement and process noise. To evaluate early-warning signals estimated from each component, trends in the signals prior to the critical transition are estimated based on the non-parametric trend statistic Kendall's $\tau$ [32]. Kendall's $\tau$ takes values between $-1$ and $1$, where $-1$ represents a monotonic negative trend, and $1$ represents a monotonic positive trend. A large positive Kendall's $\tau$ typically indicates that the system is approaching a transition (also, see [33,34] for more details regarding the interpretation of Kendall's $\tau$). The distribution of the measured Kendall's $\tau$ for the variance recorded at each node shows that species 2 and 3 provide the strongest Kendall's $\tau$ in most cases (figure 3), indicating that they are the best indicator species in this system, while species 1 and 4 did not provide reliable early-warning signals.

Next, we performed a sensitivity analysis on the estimated early-warning signals recorded from each species. We study how sensitive is Kendall's $\tau$ extracted from measurements of each species to the choice of different values of the data analysis parameters, and how robust are the identified Kendall's $\tau$ with respect to measurement noise. We calculated the Kendall's $\tau$ of the measured variance for a range of sampling resolutions and window sizes for each of the four species. The results are shown in figure 4 revealing that the Kendall's $\tau$ values recorded from measuring species 2 and 3 are smooth functions of the time resolution and the window size. In contrast, the plot for species 1 shows that the recorded Kendall's $\tau$ exhibits an erratic variation for different choices of data analysis parameters. For species 4, the plot shows a behaviour between that of species 1 and that of species 2 and 3. Next, a 3% relative Gaussian measurement noise was added to the recorded time series to analyse the robustness of the reported Kendall's $\tau$ value in figure 4 for each species. Comparing the results of this analysis (figure 5) with the previous results suggests that the reported values of Kendall's $\tau$ for species 2 and 3 are robust to measurement uncertainties. The values of Kendall's $\tau$ recorded from species 4 are significantly affected by the added measurement noise, even in regions where there are high resolution data available and a wide window is selected. Species 1 shows an erratic behaviour, as before. Thus, early-warning signals based on species 1 are not reliable.

The results of this section highlight the importance of identifying the best indicator species of regime shifts in a system to achieve more significant warning signals which are robust to the choice of data analysis parameters as well as uncertainties in measurements.

## 3. Data-driven algorithm to identify the direction of slowest dynamics in an interconnected network

In this study, data-driven techniques traditionally used for experimental modal analysis in engineering are used in combination with early-warning signals to identify the most reliable indicator species in a

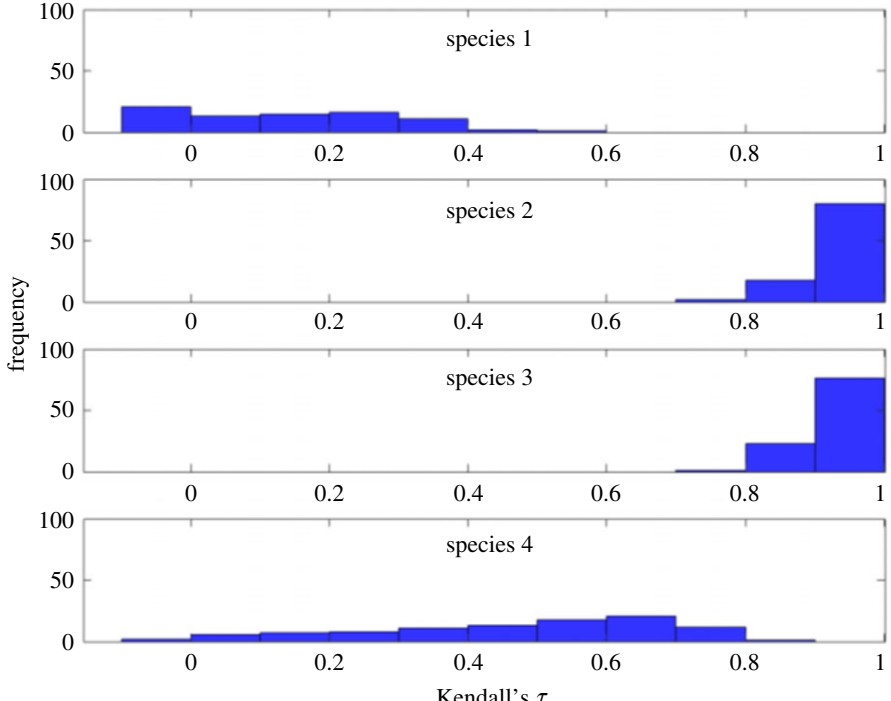

**Figure 3.** Distribution of computed Kendall's $\tau$ of early-warning signals recorded at each node for 100 independent simulations with random measurement and process noise. Species 2 and 3 are the best to monitor because they provided the most significant increasing trend in their early-warning signals.

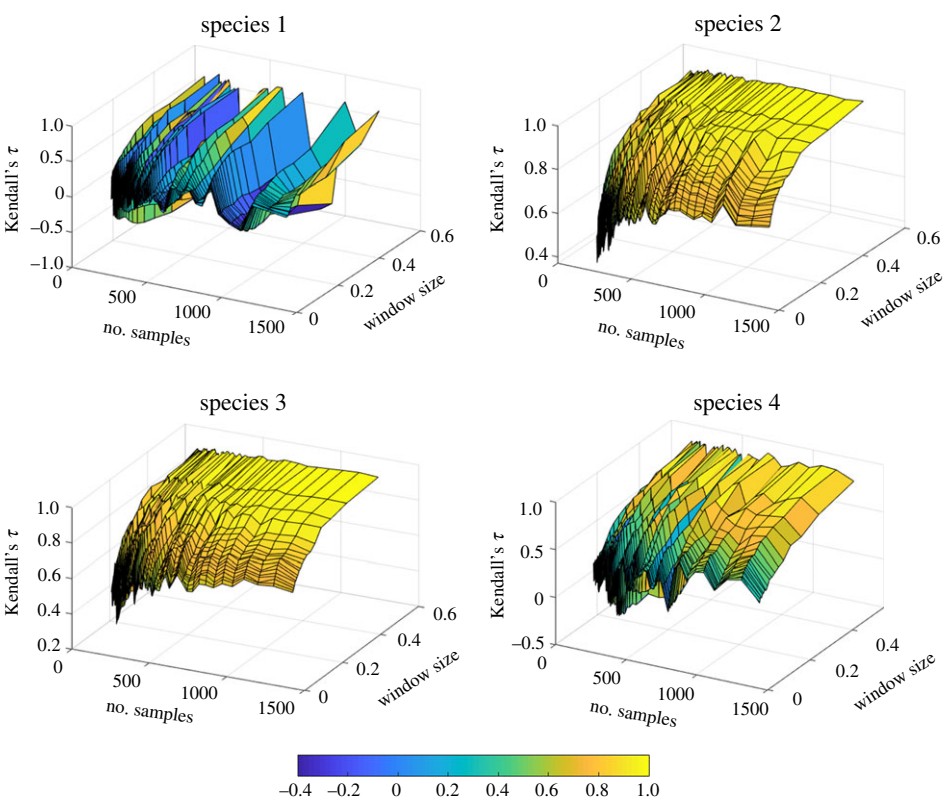

**Figure 4.** Kendall's $\tau$ values obtained using measurements of each of the four species of a system governed by equation (2.1). For a single observation of the system dynamics as it approaches the critical transition, Kendall's $\tau$ is approximated using measurements taken at each of the four species for different sampling resolutions and window sizes.

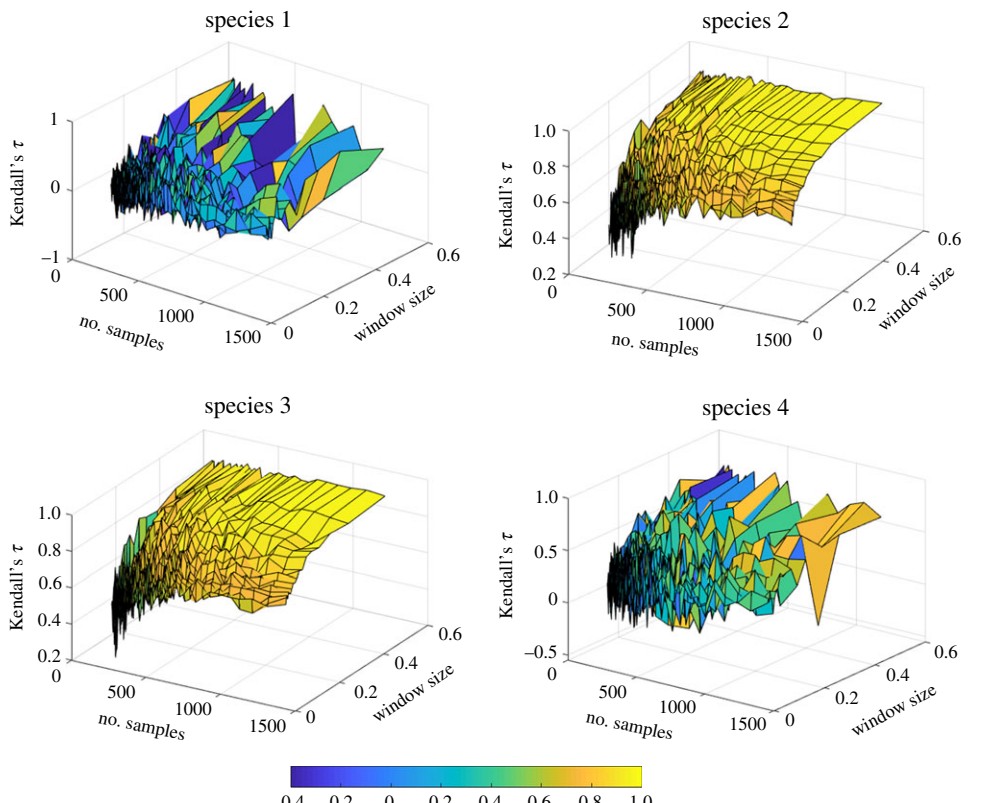

**Figure 5.** Kendall's $\tau$ values obtained for the time series used to construct figure 4 with added measurement uncertainty to each surrogate measurement data sample.

connected system for predicting critical transitions. In particular, a data-driven algorithm known as the eigensystem realization algorithm (ERA) is used, which is a method for identifying system dynamic characteristics using time-domain measurements [35]. This time-domain multi-input multi-output algorithm provides estimates of modal parameters of the system (i.e. eigenvalues and eigenvectors in the vicinity of an equilibrium) using measurements of system responses to external inputs and perturbations.

Consider a situation in which a stable system is exposed to an initial perturbation that results in a free response (e.g. a decay) to its equilibrium state. Let y denote an $n$-dimensional vector of measured state variables as the system recovers from perturbations. Further, consider that the measured response is sampled with a $\Delta t$ time increment, and $y(k)$ denotes the measured state variables at $t = k \Delta t$, $k = 0, 1, 2, \ldots,$. If the measurements are available from $m$ independent experiments, one can define $Y(k)$ as a sequence of experimentally measured state variable values as $t = k \Delta t$, i.e. $Y(k) = [y_1(k)$ $y_2(k) \ldots y_m(k)]$. The eigensystem realization algorithm is based on time-delay embedding of the measurements, a method to enrich the information that can be drawn from limited observations of a system [35,36]. The first step in the algorithm is to form generalized Hankel matrices created by the delay embedding of time series measurements on the observables [34,35],

$$H(k) = \begin{bmatrix} Y(k) & Y(k+1) & \ldots & Y(k+v-1) \\ Y(k+1) & Y(k+2) & \cdots & Y(k+v) \\ \vdots & \vdots & \ddots & \vdots \\ Y(k+r-1) & Y(k+r) & \cdots & Y(k+r+v-2) \end{bmatrix}_{nr \times mv}, \tag{3.1}$$

where parameters $r$ and $v$ are the parameters controlling the embedding dimensions and can be tuned for optimal accuracy by convergence studies [35,37]. The singular value decomposition of $H(0)$ can be written as $H(0) = \underline{P}\,\underline{Z}\,\underline{J}^T$, with the singular values in the diagonal matrix $\underline{Z}$ ordered in decreasing order (largest on the first row). The rank of the Hankel matrix is determined by the number of non-zero singular values in $\underline{Z}$. However, the presence of measurement noise and/or (weak) nonlinearities leads to additional non-zero singular values of small magnitude [38]. Selecting the first N largest singular values (i.e. the dominant ones), matrices $\underline{Z}$, $\underline{J}$ and $\underline{P}$ are truncated and denoted by Z, J and P,

respectively. Next, the state transition matrix S can be defined as [38]

$$S = Z^{-1/2} \, P^T \, H(1) \, J \, Z^{-1/2}. \tag{3.2}$$

It can be shown that the system eigenvalues are

$$\eta_i = \frac{1}{\Delta t} \ln(\tilde{s}_i) = \sigma_i \pm i\omega_d, \tag{3.3}$$

where $\tilde{s}_i$ is the $i$th eigenvalue of matrix $S$. Furthermore, matrix C contains the desired eigenvectors of systems as follows:

$$C = (E_n^T P Z^{1/2}) T, \tag{3.4}$$

where $E_n^T = [\, I_n \, O_n \, O_n \, \dots \, O_n \,]$; $I_n$ and $O_n$ represent the identity and zero matrices of order $n$, and T is the matrix of eigenvectors of S.

The slowest dynamics of the ecosystems can be estimated based on this approach without *a priori* knowledge of the underlying system equations (i.e. without a mathematical model of the system, but using only time-series measurements of some of its states). The measured free decay responses of system exposed to small or large perturbations are used to identify the slowest eigenvalue $\eta_s$ and its corresponding eigenvector $c_s$ using equations (3.1) to (3.4). Once $c_s$ is determined, it is not necessary to monitor or measure all species but the subset of species corresponding to the largest absolute values in $c_s$ which have the most participation in the slowest system dynamics where the slowing down is expected to occur.

## 3.1. Eigensystem realization using stochastic fluctuations

The algorithm presented above uses free decay response to perturbations to extract system dynamical features. For many systems, including ecological systems, to observe or measure system free decays in response to large enough perturbations may be infeasible or impossible. Small amplitude natural stochastic excitations, however, may always exist in the dynamics of these systems. It can be shown that such excitations may provide sufficient response for the identification algorithm (ERA) to be performed [37,39,40]. Particularly, one can show that the auto-correlation functions of the system response to stochastic excitations have the same characteristics as the free decay responses of the system and can be used as an input to the ERA algorithm. This results in the method referred to as the natural excitation technique (NExT) [37,39,40], which is summarized below.

Consider the following linear differential equation:

$$\dot{x} = A x + f(t), \tag{3.5}$$

where $x \,\epsilon\, R^n$ is the system state vector, $A \,\epsilon\, R^{n \times n}$ is the system matrix and $f(t) \,\epsilon\, R^n$ represents the additive effects of environmental stochasticity. The excitation and responses are assumed to be stationary random processes, and the matrix $A$ is deterministic. Post-multiplying equation (3.5) by a reference scalar response process $x_{ref}(s)$ and taking the expected value of each side yields:

$$E(\dot{x} \, x_{ref}(s)) = A E(x \, x_{ref}(s)) + E(f(t) \, x_{ref}(s)), \tag{3.6}$$

where $s = t - \tau$, and $x_{ref}(s) \,\epsilon\, R$ is the signal measured from one of the system nodes, called the reference node. Assuming f(t) to be a white noise excitation, f(t) and $x_{ref}(s)$ become uncorrelated for all values of $\tau > 0$ owing to the fact that the mean values of f(t) and $x_{ref}(s)$ are zero. Hence, one can show that the following equation holds [37,40]:

$$\frac{d}{d\tau} R_{x \, x_{ref}}(\tau) = A R_{x \, x_{ref}}(\tau), \tag{3.7}$$

where $R_{x \, x_{ref}}(.) \,\epsilon\, R^n$ denotes the vector of correlation functions between the vector of state variables x and the reference signal $x_{ref}$. Hence, the vector of correlation functions $R_{x \, x_{ref}}(\tau)$, satisfies the homogeneous differential equation of the system dynamics. The cross-correlation functions found by measuring stochastic fluctuations of system components can therefore be used in the Hankel matrices of the ERA method instead of the free decay responses.

Each of the $n$ measured signals from system can be selected as the reference signal in equation (3.7). In general, we can form a matrix containing the correlations vectors obtained from choosing different reference channels as follows:

$$\tilde{R}(\tau) = [\, R_{x x_1}(\tau) \quad R_{x x_2}(\tau) \quad \cdots \quad R_{x x_n}(\tau) \,]_{n \times n}, \tag{3.8}$$

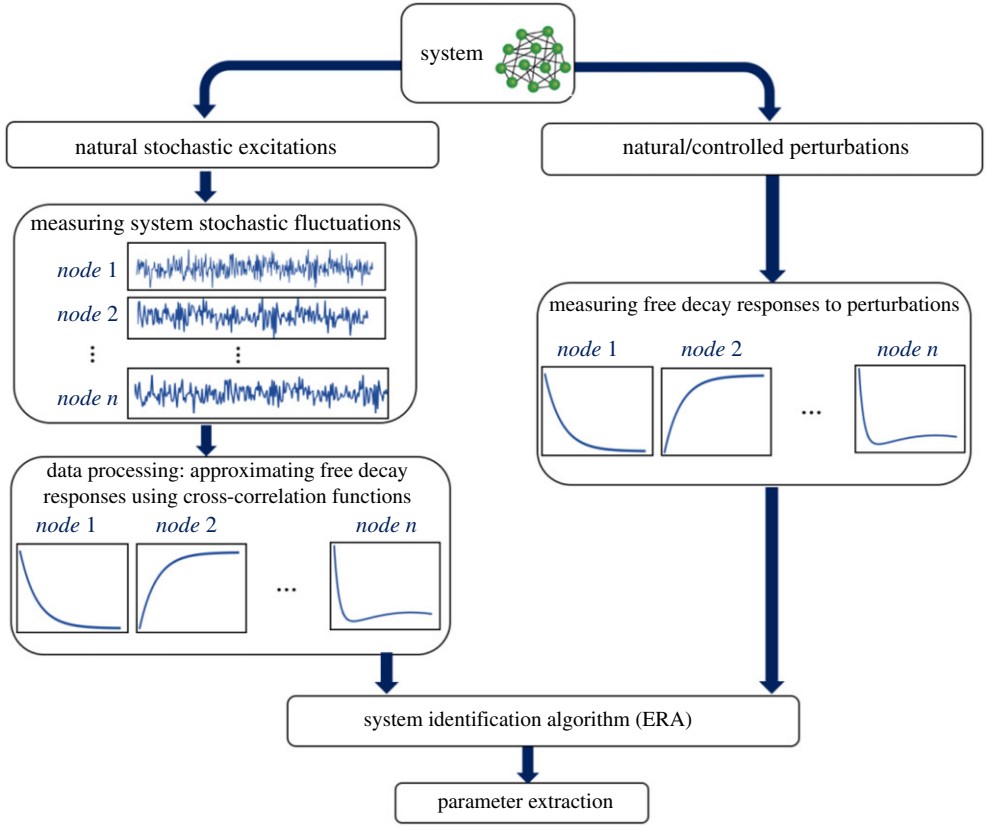

**Figure 6.** Schematic illustration of the numerical procedure. One can use both stochastic fluctuations and free decay responses from perturbations to identify dynamical properties of a system.

where $R_{xx_i}(\tau)$ denotes the vector of correlation functions between the vector of state variables x and the measured signal at location/node/component $i$. The Hankel matrices required for the ERA algorithm are constructed as

$$
H(k) = \begin{bmatrix}
\bar{R}(k) & \bar{R}(k+1) & \ldots & \bar{R}(k+v-1) \\
\bar{R}(k+1) & \bar{R}(k+2) & \cdots & \bar{R}(k+v) \\
\vdots & \vdots & \ddots & \vdots \\
\bar{R}(k+r-1) & \bar{R}(k+r) & \cdots & \bar{R}(k+r+v-2)
\end{bmatrix},
\tag{3.9}
$$

where $\tau = k\,\Delta t$, $k = 0, 1, 2, \ldots$, and $\bar{R}(k) = \tilde{R}(k\Delta t)$. Using the Hankel matrix in equation (3.9), the rest of the procedure is the same as described in equations (3.2) to (3.3). Note that one does not need to form the matrix $\tilde{R}(\tau)$ using all possible reference choices. Using only a small subset of reference signals suffices to obtain the desired outcomes provided that the selected references are not simultaneously zero for any desired system eigenvector (particularly, the slowest eigenvector in this study). Increasing the number of references and experiments, though, decreases the risk of missing information in the analysis procedure. A schematic of the methods and procedures described in this section is depicted in figure 6.

# 4. Results

In the following examples, the data-driven approach presented above is employed to rank the components of example ecological systems based on their reliability in providing valid early-warning signals of upcoming transitions. The goal is to identify which subset of species is to be monitored for the rest of the system life to detect the upcoming transition reliably. It is assumed that the available measurements are fluctuations in system dynamics in response to natural stochastic external fluctuations, and the system is at a parameter value before the critical transition.

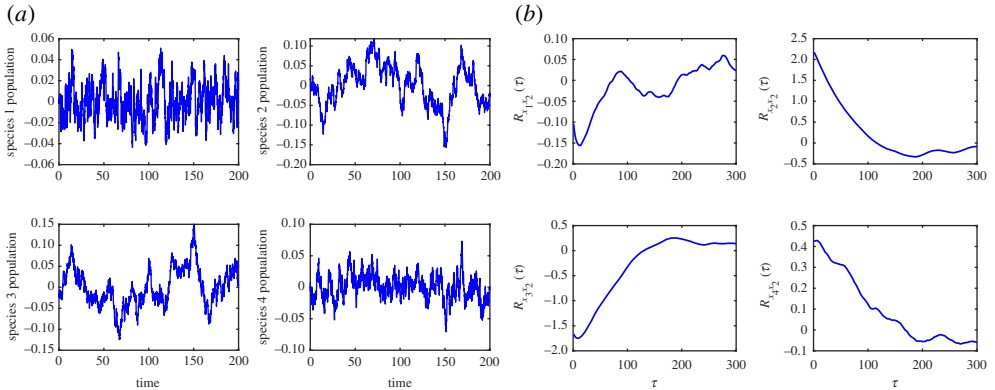

**Figure 7.** (*a*) Measured stochastic system fluctuations around the equilibrium at $\mu = 4.5$ used to identify the slowest direction of a network of four interacting species. (*b*) Example of approximated free decays of the cross-correlation functions (reference channel: species 2) used as an input to the ERA algorithm.

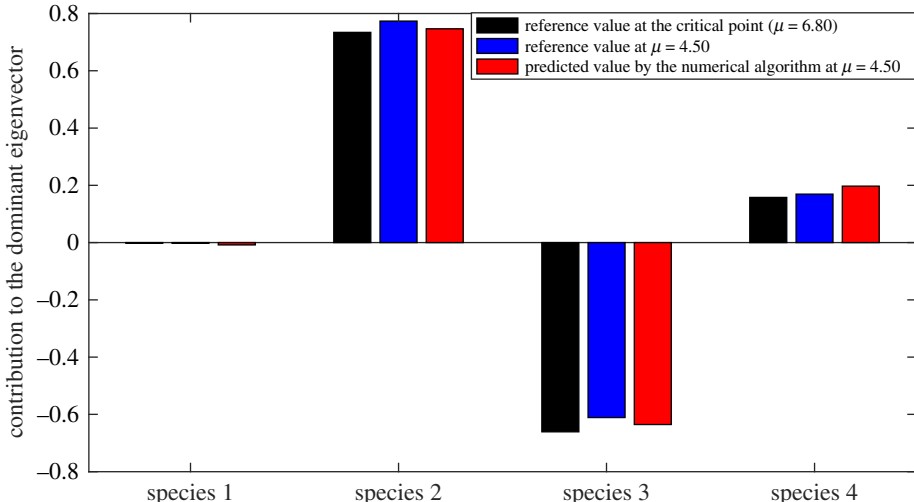

**Figure 8.** Slowest eigenvector of the system estimated using the proposed approach and the exact eigenvector. The data-driven approach correctly identifies nodes 2 and 3 as the ones with the most contribution and node 1 as the one with the least contribution to the slowest eigenvector.

## 4.1. Network of four interacting species

In the first example, we considered the network of four connected species studied in §2 (figure 1). The goal of the analysis is to rank the species of this system (i.e. a ranking of the nodes) based on their reliability in providing early-warning signals of the upcoming critical transition using the proposed data-driven method. Theoretical analysis of the equations shows that the exact dominant eigenvector of the system at the critical transition is $v_c = [-0.002, 0.734, -0.661, 0.158]^T$, reflecting that species 2 and 3 are the best ones to be monitored for identifying the upcoming transition. By contrast, species 1 and 4 participate the least in the dominant eigenvector. Thus, if the system stability were evaluated by monitoring species 1 and/or 4, then only a weak signal is detected until the system is extremely close to the transition. To identify the best indicator species for this system without a model of the system, we applied the data-driven algorithm to identify the slowest eigenvector of the system using measured system recoveries to random stochastic excitations at $\mu = 4.5$ (figure 7).

Figure 8 shows a comparison of the exact dominant eigenvector computed using the theoretical formulation and the approximated eigenvector obtained by the data-driven algorithm. The results suggest that the best indicator species can be identified using this model-free, data-driven approach. The algorithm successfully ranked the species based on their importance regarding the extraction of reliable early-warning signals using measurements of the system response to perturbations. The algorithm correctly ranks species 2 as the best indicator species and species 3 as the second-most

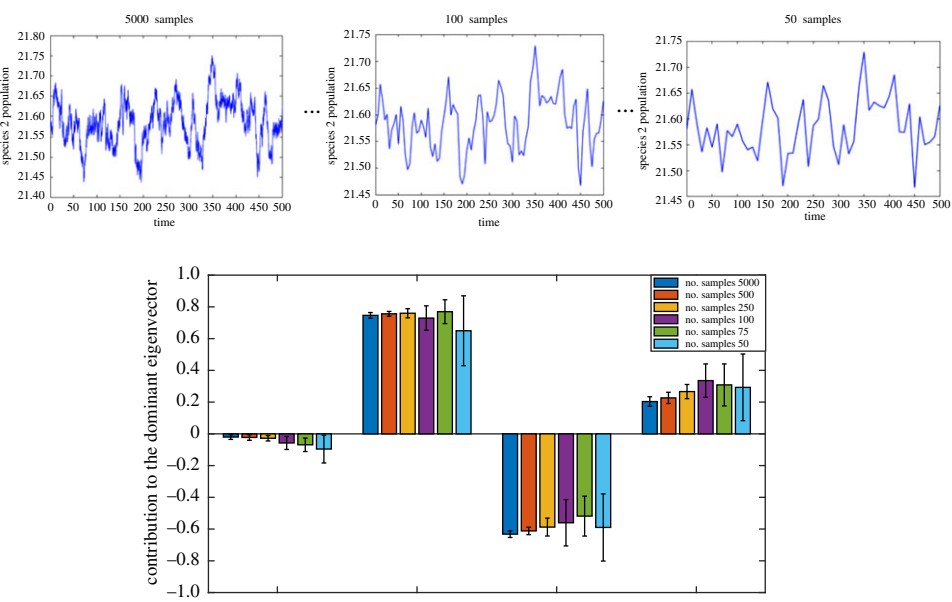

**Figure 9.** Sample size effect on identification of the dominant eigenvector using the proposed approach.

reliable option, while it indicated that measuring species 1 would not provide a strong signal of approaching an upcoming critical transition.

Similar results are obtained when the system free decay responses to a large perturbation is used as an input to the algorithm (results not shown due to the similarity). The advantage of applying control large perturbations is that a relatively short measurement suffices to approximate system dynamics while in the case of stochastic excitations the measured time series should be long enough so that the approximated correlation-based free decays have acceptable accuracy.

For many systems, including ecological and biological systems, to collect a large sample size from system is challenging. We performed an analysis to investigate the sample size effect on the approximation of the slowest eigenvector of the system. It is assumed that the monitored period is 500 time units (arbitrary time units that can be hours or minutes, for example), and we vary the sampling rate to decrease the number of inputs to the presented eigenrealization algorithm. Results of this analysis are demonstrated in figure 9. As expected, it is observed that the accuracy is decreased as a result of a decrease in sampling size. However, even a small number of samples results in an approximation of the eigenvector that is good for ordering the nodes.

## 4.2. Spatial harvesting model

In this section, we considered a two-dimensional spatially distributed ecosystem, namely a harvesting model. Identifying the best species to monitor in a system of high dimensions, or the area to monitor in a spatially distributed system, is of particular importance. When the system is spatially distributed and has a large dimension owing to a large number of interacting regions, it is costly and often infeasible to monitor the dynamics of the whole system. Therefore, it is desirable to identify a subset of indicator regions that can provide reliable early-warning signals of upcoming critical transitions. Such indicator regions are viewed similar to indicator species in the previous examples.

The harvesting model can be discretized in a grid of interconnected regions to obtain the model

$$\frac{dx_{i,j}}{dt} = r_{i,j}x_{i,j}\left(1 - \frac{x_{i,j}}{K_{i,j}}\right) - c_{i,j}\frac{x_{i,j}^2}{x_{i,j}^2 + 1} + D(x_{i+1,j} + x_{i-1,j} + x_{i,j+1} + x_{i,j-1} - 4x_{i,j}) + \sigma_{i,j}dW_{i,j}, \qquad (3.10)$$

where $x_{i,j}$ is the biomass in region $(i, j)$ (i.e. a scalar state variable for each region), $K_{i,j}$ is the carrying capacity in region $(i, j)$, $c_{i,j}$ is the harvesting rate in region $(i, j)$, $D$ is the dispersion rate, $r_{i,j}$ is the maximum growth rate in region $(i, j)$, and $\sigma_{i,j}$ is the standard deviation of the noise excitation in region $(i, j)$, for $i = 1, \ldots, P$, and $j = 1, \ldots, Q$, with $P$ and $Q$ being the number of nodes in the two dimensions of the spatially distributed system. The dynamics in each region $(i, j)$ are affected by a reaction process described by the first term on the right-hand side in equation (3.10), namely the

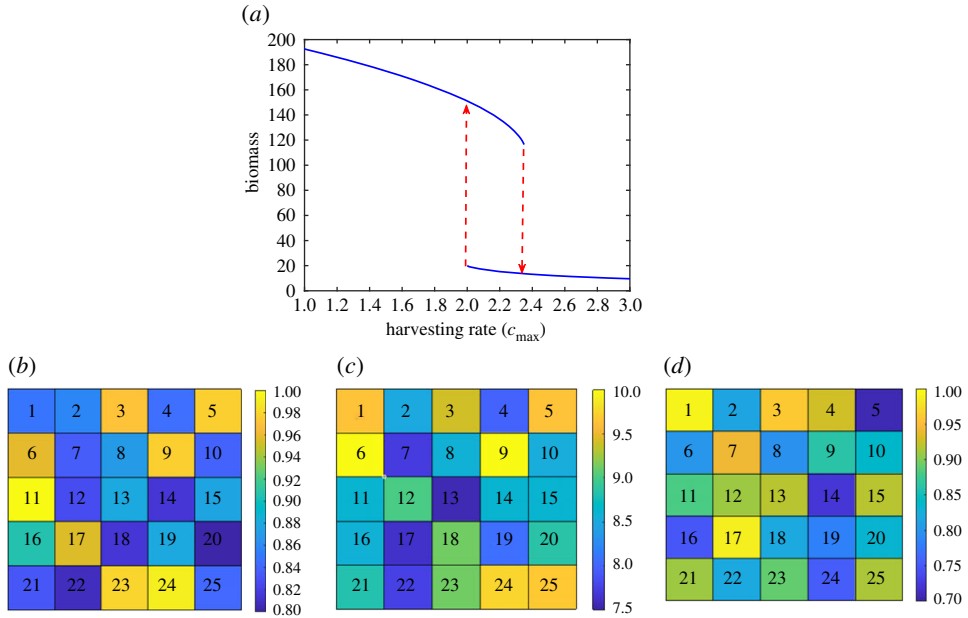

**Figure 10.** (a) Bifurcation diagram of the spatial harvesting model. Only the stable branches are shown in this figure. (b) The value of the growth rate $r_{i,j}$ in each region. (b) The value of the carrying capacity $K_{i,j}$ in each region. (d) The value of the harvesting rate $\mu_{i,j}$ in each region.

nonlinear deterministic term $r_{i,j}x_{i,j}\left(1-(x_{i,j}/K)\right) - c_{i,j}(x_{i,j}^2/x_{i,j}^2 + 1)$. Each region also interacts with its neighbouring regions (with periodic boundary conditions) through a diffusion process with dispersion rate $D$. We assume that independent random excitations exist in each region, represented by independent random walk processes $dW_{i,j}$. We define the harvesting rate in region $(i, j)$ as $c_{i,j} = \mu_{i,j}c_{\max}$, where $\mu_{i,j} \in [0.7, 1]$, meaning that the harvesting rate is non-uniform among different patches. By increasing the harvesting rate $c_{\max}$, the system exhibits a critical transition from under-exploitation (a high-population equilibrium) to over-exploitation (a low-population equilibrium).

The proposed data-driven approach was applied to identify the best indicator regions in this large-dimensional ecosystem. Considering $P = Q = 5$ results in a 25-dimensional system. The bifurcation diagram of the system is depicted in figure 10a. The dispersion rate was selected to be $D = 0.2$ for all patches. To introduce spatial heterogeneity to the system, the growth rates $r_{i,j}$, carrying capacities $K_{i,j}$ and harvesting rates $\mu_{i,j}$ of each region were selected randomly from the intervals [0.8,1], [7.5,10] and [0.7,1], respectively (figure 10).

Identifying the best indicator regions in this system is important because monitoring all regions would require an excessive cost and effort. Thus, the proposed data-driven approach was employed with measured system responses recorded at a harvesting rate of $c_{\max} = 2$. Theoretical analysis of the system shows that the slowest eigenvector of the system is that shown in figure 11a. Thus, the best indicator regions for this system are those around the bottom left corner of the field. The results of this model-free, data-driven approach are shown in figure 11b, and are consistent with the theoretical predictions. These results indicate that the most important regions in this spatially distributed system can be accurately estimated without any knowledge of the underlying system equations. Region 21 was identified as the most-favourable region to be monitored, while the regions at the top right corner were correctly identified as the least-favourable regions to be monitored.

In spatially distributed systems, it might be challenging to measure the entire system to approximate the dominant eigenvector. Furthermore, even if the best indicator region is known, it might not be accessible or feasible before long-term measurements to extract early-warning signals for critical transitions. Hence, it is important to identify the best indicator regions among those that are accessible for measurement. In the next example, we assume that only some of the regions are accessible for measurements in the spatial harvesting model. For instance, we assume that the regions in the bottom and left edges of the system are the only measurable ones, and their response to small random perturbations is employed in the data-driven algorithm. Figure 11c shows the ranking result for the measured patches. Results of this data-driven approach correctly detect that region 21 is the best indicator region compared with the other measured regions.

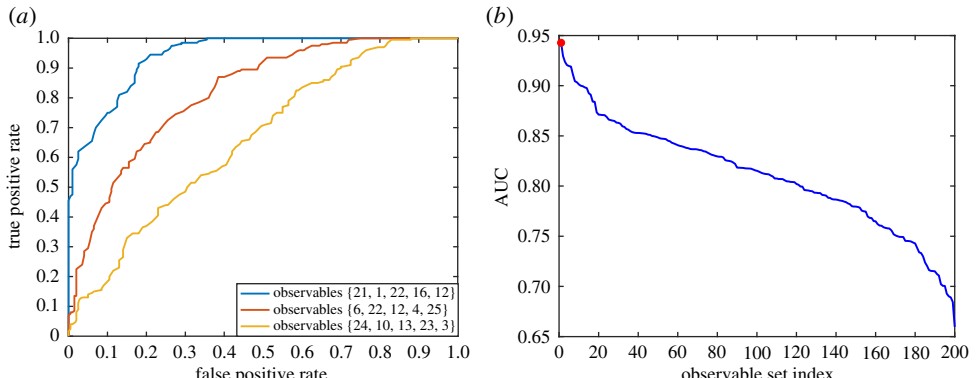

**Figure 11.** The slowest eigenvector obtained using (*a*) the theoretical formulation, (*b*) the data-driven algorithm by measuring all regions and (*c*) the data-driven algorithm measuring only the regions along the left and bottom edges of the two-dimensional domain. Regions that were not measured are shown in white; no information is available for these regions. Results from all analyses suggest that region 21 is the best indicator region.

**Figure 12.** (*a*) Example of ROC curve for three different sets of observables obtained from 400 independent simulations. (*b*) AUC statistics showing the performance of the randomly selected patches. Sets are indexed based on their AUC score. The first index corresponds to the observables selected based on identified dominant eigenvector of the dynamics, i.e. {1, 12, 16, 21, 22}, which resulted in the highest AUC score.

To explore the effectiveness of sampling based on the proposed approach, we analysed ROC and AUC statistics of early-warning signals obtained from different choices of observables given a certain sampling effort. Particularly, in the harvesting model presented in equation (3.10), it is assumed that only five patches can be selected and monitored to probe for the signals of potential upcoming transitions. Four hundred independent simulations with random initial control parameters ($c_{max}$) were performed, with only half of them approaching the critical point. We randomly select 200 sets of observables, each set includes 5-permutation of the 25 patches. For each set of observables, we compute the Kendall's $\tau$ of the warning signals obtained from aggregating the population of the patches. The ROC curves and the AUC statistics are then constructed for each set of observables by varying Kendall's $\tau$ from 0 to 1 as a binary threshold above which an upcoming transition is alarmed. The results are compared to the case that observables are selected based on their contribution to the identified eigenvector shown in figure 11, i.e. patches {21, 1, 22, 16, 12}. Results of this analysis are demonstrated in figure 12, showing that the set chosen based on the dominant eigenvector outperforms the randomly selected sets by providing the strongest warning signal of the upcoming transition.

# 5. Discussion and conclusion

Detecting early-warning signals of critical transitions for large-dimensional complex ecosystems is a challenging task because of the complex system dynamics involving many species. Changes in the system affect different species in different ways and may have much larger impacts on the dynamics of some species than others. We investigated examples highlighting that not all the species/nodes/regions in an ecosystem provide reliable early-warning signals of a system approaching a transition. In addition, early-warning signals measured from dynamics of specific system components were shown to be robust

to the choice of data analysis parameters as well as uncertainties in measurements making them reliable components to be monitored, while the warning signals measured from dynamics of some nodes are not reliable. We discussed why such a group of nodes exist in a system and validated the effect of the choice of observers on the extracted early-warning indicators. According to the slowing-down phenomenon, system states/nodes with the highest contribution to the dominant eigenvector (the slowest direction) of the system dynamics are in fact the subgroup of interest to be monitored in a network.

In complex dynamical systems, to develop model-free methods determining where to measure the system and how to interpret the measurements to achieve the most accurate early-warning signals of an upcoming transition is an important requirement. Herein, we proposed a data-driven approach to rank the species/nodes/regions of a system based on their reliability in providing the best early-warning signals of critical transitions. We used experimental modal analysis techniques, typically used in structural dynamics in engineering, combined with early-warning signals to identify the best-indicator species/nodes/regions in a connected system and extracted the most reliable early-warning signals of critical transitions. The system response to environmental excitations and natural/control perturbations were recorded and used as an input for the proposed technique, which is a tool to identify dynamical features. Numerical examples demonstrated that the proposed technique was able to successfully rank the species/nodes/regions in the studied ecosystems based on the accuracy of their provided early-warning signals.

Assuming that system dynamics and properties change smoothly over time, the approximated ranking remains valid for a reasonable period of time. Thus, the identification process might be required to be updated only a few times, if at all, during the life cycle of the system. Hence, although the identification procedure might initially require extra measurement and data processing efforts for a short period of time, this effort results in a reliable stability analysis of the system and reduces measurement cost and efforts afterwards. Note that even if one is able to measure all system components, information collected from some of the nodes might be misleading and cause errors. Identification of the dominant slow direction is of great importance for interpretation and post-processing of the measured signals.

Despite the advantages of the proposed algorithm and similar studies which rely on the slowest direction in the system to identify the risk of an impending transition, one should consider other factors which affect the interpretation of the results or even challenge the validity of the results. First, one should note that relying on the slowest direction of the dynamics for identifying both the impending transitions and reliable sensor species holds only when the slowest eigenvector is causing instability in the system. The main assumption behind slowing-down-based methods is that the system is close enough to the transition so that the slowest (dominant) eigenvector is the most likely one which is going to make the system unstable. In some systems, it might be possible that one of the eigenvectors, which is not the slowest one for most of the system life, suddenly loses its stability and drives the system toward instability. In such cases, approximated early-warning signals and node ranking results are dominated by the slowest dynamics and cannot capture the upcoming transition unless the system is significantly close to the transition. To detect such a condition, one needs to frequently monitor the trend of all the system eigenvalues and eigenvectors, which is a demanding task. The proposed numerical algorithms in this paper allow one to identify a wide range of system eigenvalues and eigenvectors, not just the slowest one. Hence, this makes the proposed methods a good candidate to monitor substantial changes in the system dynamics. However, performing such an analysis requires permanent costly measurement of a large subset of system components, accurate and high-frequency measurements, and detailed post-processing of the data and results. Due to the uncertainty and limitations in the measurements of real complex systems, this task can be quite challenging, although theoretically possible.

In large-dimensional systems, the effects of several eigenvectors other than the dominant one might need to be considered to successfully rank the importance of the species/nodes/regions in the system. In large-dimensional systems, the system has a large number of eigenvalues. Several of these eigenvalues other than the dominant one might slow down and approach zero as the system approaches the transition. Hence, there may be several eigenvalues slowing down and affecting the early-warning signals, particularly those with a close value to the dominant eigenvalue. In such cases, the system state that contributes the most to the dominant eigenvector would still be among the best to be measured. However, small entries in the dominant eigenvector do not necessarily mean that their corresponding states do not provide useful early-warning signals. Hence, the focus of node ranking in a large network should be to identify the reliable nodes to be monitored.

Although the focus of this paper is to identify the slowest direction of the dynamics with the goal of identifying the reliable sensor species to predict critical transition, the approach presented also unveils

detailed information about the underlying system dynamics. Particularly, the proposed method allows one to identify not only the slowest direction of the dynamics but also a wide range of other system eigenvectors and their corresponding eigenvalues all at once. Such detailed information about the underlying system dynamics is valuable for a variety of purposes, e.g. identifying the most and the least resilient direction of a network which is required for managing the system dynamics, and post-processing measured system responses by projecting them on directions of interest to enhance the analysis of system changes in multiple subspaces.

The result of this study addresses one of the important challenges in stability analysis of dynamical networks by introducing a new theoretical framework. The methods and ideas presented here can be used in developing practical tools to analyse the stability and resilience of dynamical systems, ranging from ecological and biological to engineered systems.

Data accessibility. The codes developed based on the methods described in this article are archived in the Dryad Digital Repository: https://doi.org/10.5061/dryad.zgmsbcc7x [41].

Authors' contributions. A.G. and S.C. performed the simulations. A.G., S.C and B.I.E jointly developed the algorithms, interpreted the results and wrote the paper.

Competing interests. We declare we have no competing interests.

Funding. This research was supported by the National Institute of General Medical Sciences of the National Institutes of Health under Award no. U01GM110744. The content is solely the responsibility of the authors and does not necessarily reflect the official views of the National Institutes of Health.

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
