## [Reviewer comments · Royal Society Open Science]

Review History

RSOS-200896.R0 (Original submission)

Review form: Reviewer 1

Is the manuscript scientifically sound in its present form?

Yes

Are the interpretations and conclusions justified by the results?

Yes

Is the language acceptable?

Yes

Do you have any ethical concerns with this paper?

No

Have you any concerns about statistical analyses in this paper?

No

Recommendation?

Accept with minor revision (please list in comments)

Comments to the Author(s)

The manuscript has been significantly improved. Here are a few comments:

1. Line 14: the variable r_i is not shown in eq.(1).
2. Line 39: "small measurement uncertainty " is not clear. Section 2 is too long, but it only shows an example to the problem.
3. It seems that the ERA is identical to the procedure of finding the DNB group (the simplest case $r = v = 1$ in Hankel matrix (2)). When $r, v > 1$, the author needs to describe it or compare the ERA with the DNB.

Review form: Reviewer 2

Is the manuscript scientifically sound in its present form?

Yes

Are the interpretations and conclusions justified by the results?

Yes

Is the language acceptable?

Yes

Do you have any ethical concerns with this paper?

No

Have you any concerns about statistical analyses in this paper?

No

Recommendation?

Major revision is needed (please make suggestions in comments)

Comments to the Author(s)

The revision has substantially improved the paper. However, I still have some remaining concerns. The eigenvalue/eigenvector estimation that is used is acceptable, although the use of van Kampen's formula (for example Barter's paper <https://arxiv.org/abs/1910.09698>) seems to solve the same problem in a more principled way.

The new and exciting idea is that one perhaps needs to do this data-intensive analysis only once, and can then use the insights gained to construct much simpler warning signals. However, I am still not convinced that this is actually true. It seems that there is a significant risk that transitions are overlooked. This needs to be addressed either statistically by showing that following the authors proposed method one gets a better ROC statistics given a certain sampling effort, or by further refinement of the approach that reduces this risk. (Actually using multiple eigenvalues, how many? How chosen? Or, perhaps rescaling the eigenvalues/eigenvectors by the turnover rates of the species involved. To compensate for allometric scaling)

Decision letter (RSOS-200896.R0)

Dear Dr Ghadami,

The editors assigned to your paper ("Data-Driven Identification of Reliable Sensor Species to Predict Regime Shifts in Ecological Networks") have now received comments from reviewers.

We would like you to revise your paper in accordance with the referee and Associate Editor suggestions which can be found below (not including confidential reports to the Editor). Please note this decision does not guarantee eventual acceptance.

Please submit a copy of your revised paper before 10-Jul-2020. Please note that the revision deadline will expire at 00.00am on this date. If we do not hear from you within this time then it will be assumed that the paper has been withdrawn. In exceptional circumstances, extensions may be possible if agreed with the Editorial Office in advance. We do not allow multiple rounds of revision so we urge you to make every effort to fully address all of the comments at this stage. If deemed necessary by the Editors, your manuscript will be sent back to one or more of the original reviewers for assessment. If the original reviewers are not available, we may invite new reviewers.

- Data accessibility

If you wish to submit your supporting data or code to Dryad (<http://datadryad.org/>), or modify your current submission to dryad, please use the following link:
<http://datadryad.org/submit?journalID=RSOS&manu=RSOS-200896>

- **Competing interests**

- **Authors' contributions**

- **Acknowledgements**

- **Funding statement**

Kind regards,

Lianne Parkhous

Editorial Coordinator

on behalf of the Associate Editor and Professor Pete Smith (Subject Editor)

Associate Editor's comments to the Author:

Thank you for your efforts to respond to the reviewers' queries. As you'll see, they are largely persuaded by the changes made; however, one of the reviewers has a number of outstanding queries that need to be addressed in greater depth. We'll look forward to receiving the revised submission in the near future.

Reviewers' Comments to Author:

Reviewer: 1

Comments to the Author(s)

The manuscript has been significantly improved. Here are a few comments:

1. Line 14: the variable r_i is not shown in eq.(1).
2. Line 39: "small measurement uncertainty " is not clear. Section 2 is too long, but it only shows an example to the problem.
3. It seems that the ERA is identical to the procedure of finding the DNB group (the simplest case $r = v = 1$ in Hankel matrix (2)). When $r, v > 1$, the author needs to describe it or compare the ERA with the DNB.

Reviewer: 2

Comments to the Author(s)

The revision has substantially improved the paper. However, I still have some remaining concerns. The eigenvalue/eigenvector estimation that is used is acceptable, although the use of van Kampen's formula (for example Barter's paper <https://arxiv.org/abs/1910.09698>) seems to solve the same problem in a more principled way.

The new and exciting idea is that one perhaps needs to do this data-intensive analysis only once, and can then use the insights gained to construct much simpler warning signals. However, I am still not convinced that this is actually true. It seems that there is a significant risk that transitions are overlooked. This needs to be addressed either statistically by showing that following the authors proposed method one gets a better ROC statistics given a certain sampling effort, or by further refinement of the approach that reduces this risk. (Actually using multiple eigenvalues, how many? How chosen? Or, perhaps rescaling the eigenvalues/eigenvectors by the turnover rates of the species involved. To compensate for allometric scaling)

Author's Response to Decision Letter for (RSOS-200896.R0)

See Appendix A.

RSOS-200896.R1 (Revision)

Review form: Reviewer 2

Is the manuscript scientifically sound in its present form?

Yes

Are the interpretations and conclusions justified by the results?

Yes

Is the language acceptable?

Yes

Do you have any ethical concerns with this paper?

No

Have you any concerns about statistical analyses in this paper?

No

Recommendation?

Accept as is

Comments to the Author(s)

The revision has addressed my previous concerns. I recommend publication.

Decision letter (RSOS-200896.R1)

Dear Dr Ghadami,

It is a pleasure to accept your manuscript entitled "Data-Driven Identification of Reliable Sensor Species to Predict Regime Shifts in Ecological Networks" in its current form for publication in Royal Society Open Science. The comments of the reviewer(s) who reviewed your manuscript are included at the foot of this letter.

Please ensure that you send to the editorial office an editable version of your accepted manuscript, and individual files for each figure and table included in your manuscript. You can send these in a zip folder if more convenient. Failure to provide these files may delay the processing of your proof.

on behalf of Prof Pete Smith (Subject Editor)
openscience@royalsociety.org

Associate Editor Comments to Author:

Thank you for constructively engaging with the reviewer's comments: they now recommend acceptance - congratulations and thank you for supporting RSOS!

Reviewer comments to Author:

Reviewer: 2

Comments to the Author(s)

The revision has addressed my previous concerns. I recommend publication.

Appendix A

To: **Andrew Dunn**
Senior Publishing Editor, Journal of the Royal Society Open Science
Re: Manuscript ID: RSOS-200896

July 8, 2020

Dear Dr. Dunn,

We would like to thank you and the reviewers for the feedback. The comments of the reviewers have been very helpful in improving our paper in terms of content and readability, for which we are deeply grateful. We have revised our paper and addressed the reviewers' comments. In particular, below are details regarding some of our specific corrections and revisions we made (*shown in green*) in response to reviewers' comments.

Sincerely,
the authors

Reviewer: 1

Comments to the Author(s)

The manuscript has been significantly improved. Here are a few comments:

1. Line 14: the variable r_i is not shown in eq.(1).

We fixed this error in the revised text.

2. Line 39: "small measurement uncertainty " is not clear.

We clarified the meaning of the small measurement uncertainty, which is 3% relative Gaussian measurement noise.

Section 2 is too long, but it only shows an example to the problem.

Indeed, this introductory section was too long. We removed some of the text including redundant information to shorten this section. However, we did not remove it entirely because a detailed analysis applied to an example provides clarity for many readers.

3. It seems that the ERA is identical to the procedure of finding the DNB group (the simplest case $r = v = 1$ in Hankel matrix (2)). When $r, v > 1$, the author needs to describe it or compare the ERA with the DNB.

We revised the text to clarify the ideas behind ERA and embedded coordinates, and we added appropriate references in the revised text. Specifically, parameters r and v come from the concept of embedded coordinates and embedding theorem which is the basis of the ERA approach [1,2]. Based on the embedding theorem, it is possible to enrich a measurement $x(t)$ obtained from a limited number of observations with time-shifted copies of itself $x(t - \tau)$, known also as delay coordinates. The Hankel matrix is created by the delay embedding of time series measurements on the observables, where r and v are the parameters controlling the embedding dimensions [2,3] (choosing $r = v = 1$ is not an embedding which is a requirement for this algorithm, and will not result in acceptable approximations of the system dynamics). Taking the singular value decomposition (SVD) of the Hankel matrix yields a hierarchical decomposition of the matrix into eigen-time-delay coordinates. The ERA method shows that it is possible to use the SVD of the Hankel matrix to identify accurately the underlying dynamical features of the measured systems as describe in the main text. More details of the ideas and corresponding proofs can be found in reference [2].

Reviewer: 2

Comments to the Author(s)

The revision has substantially improved the paper. However, I still have some remaining concerns. The eigenvalue/eigenvector estimation that is used is acceptable, although the use of van Kampen's formula (for example Barter's paper [//arxiv.org/abs/1910.09698](https://arxiv.org/abs/1910.09698)) seems to solve the same problem in a more principled way.

Indeed, each eigenvalue/eigenvector estimation method has its own advantages and disadvantages, which differ for each application. ERA has a long tradition in engineering applications as an effective, purely data-driven method, which approximates the system dynamics from just input-output data of a given system, regardless of availability additional information about the system (e.g. network structure, noise source and intensity). Of course, other methods exist that result in similar outcomes. Each of these methods has its own assumptions and potential advantages over other methods for specific applications. In this study, however, we selected the ERA method so that the proposed algorithms and analyses are data-driven and not system specific, which has the potential to make the approach of broadest interest across disciplines.

The new and exciting idea is that one perhaps needs to do this data-intensive analysis only once, and can then use the insights gained to construct much simpler warning signals. However, I am still not convinced that this is actually true. It seems that there is a significant risk that transitions are overlooked. This needs to be addressed either statistically by showing that following the authors proposed method one gets a better ROC statistics given a certain sampling effort, or by further refinement of the approach that reduces this risk. (Actually using multiple eigenvalues, how many? How chosen? Or, perhaps rescaling the eigenvalues/eigenvectors by the turnover rates of the species involved. To compensate for allometric scaling)

We agree that there is a risk that transitions can be overlooked when only a subset of the system is monitored. We present a method to reduce this risk by effectively choosing the system observables if there are limitations in sampling the whole system. This approach is useful because not all systems are observed everywhere at all times because of cost and effort considerations. There are cases where only a subset of the system can be observed over long times.

To explore the effectiveness of sampling based on the proposed approach, we analyzed ROC and AUC statistics of early warning signals obtained from different choices of observables given a certain sampling effort. Particularly, in the harvesting model presented in Section 4.2, it is assumed that only 5 patches can be selected and monitored to probe for the signals of potential upcoming transitions. 400 independent simulations with random initial control parameters (c_{max}) were performed, with only half of them approaching the critical point. We randomly selected 200 sets of observables, each set includes 5-permutation of the 25 patches. For each set of observables, we compute the Kendall's τ of the warning signals obtained from aggregating the population of the patches. The ROC curves and AUC statistics are then constructed for each set of

observables by varying the Kendall's τ from 0 to 1 as a binary threshold above which an upcoming transition is alarmed. The results are compared to the case that observables are selected based on their contribution to the identified eigenvector shown in Fig. 11, i.e. patches $\{21, 1, 22, 16, 12\}$. Results of this analysis are demonstrated in Fig. 12, showing that the set chosen based on the dominant eigenvector outperforms the randomly selected sets by providing the strongest warning signal of the upcoming transition.

This analysis has been added to the revised text.

(a) (b)
 Figure 12. (a) Example of ROC curves for three selected sets of observables obtained from 400 independent simulations. (b) AUC statistics showing the performance of the randomly selected patches. The sets are indexed based on their AUC score. The first index corresponds to the observables selected based on identified dominant eigenvector of the dynamics, i.e. $\{1,12,16,21,22\}$, which resulted in the highest AUC score.

References

- [1] F. Takens, Detecting strange attractors in turbulence, in: Dyn. Syst. Turbul. Warwick 1980, Springer, 1981: pp. 366–381.
- [2] J.-N. Juang, R.S. Pappa, An eigensystem realization algorithm for modal parameter identification and model reduction, J. Guid. 8 (1985) 620–627.
- [3] J.M. Caicedo, S.J. Dyke, E.A. Johnson, Natural excitation technique and eigensystem realization algorithm for phase I of the IASC-ASCE benchmark problem: Simulated data, J. Eng. Mech. 130 (2004) 49–60.